# Steering Distortions to Preserve Classes and Neighbors in Supervised Dimensionality Reduction

**Benoît Colange**
Univ. Grenoble Alpes, INES,
F-73375, Le Bourget du Lac, France
`benoit.colange@cea.fr`

**Jaakko Peltonen**
Tampere University, Faculty of Information Technology
and Communication Sciences, Finland
`jaakko.peltonen@tuni.fi`

**Michaël Aupetit**
Qatar Computing Research Institute,
Hamad bin Khalifa University, Doha, Qatar
`maupetit@hbku.edu.qa`

**Denys Dutykh**
Univ. Grenoble Alpes, Univ. Savoie Mont Blanc,
CNRS, LAMA, 73000 Chambéry, France
`denys.dutykh@univ-smb.fr`

**Sylvain Lespinats**
Univ. Grenoble Alpes, INES,
F-73375, Le Bourget du Lac France
`sylvain.lespinats@cea.fr`

## Abstract

Nonlinear dimensionality reduction of high-dimensional data is challenging as the low-dimensional embedding will necessarily contain distortions, and it can be hard to determine which distortions are the most important to avoid. When annotation of data into known relevant classes is available, it can be used to guide the embedding to avoid distortions that worsen class separation. The supervised mapping method introduced in the present paper, called ClassNeRV, proposes an original stress function that takes class annotation into account and evaluates embedding quality both in terms of false neighbors and missed neighbors. ClassNeRV shares the theoretical framework of a family of methods descending from Stochastic Neighbor Embedding (SNE). Our approach has a key advantage over previous ones: in the literature supervised methods often emphasize class separation at the price of distorting the data neighbors' structure; conversely, unsupervised methods provide better preservation of structure at the price of often mixing classes. Experiments show that ClassNeRV can preserve both neighbor structure and class separation, outperforming nine state of the art alternatives.

## 1   Introduction

Dimensionality Reduction (DR) methods aim at mapping a high dimensional dataset as points in a lower dimensional embedding space, while preserving some similarity measure between data points. DR may be supervised by taking advantage of class information. Hence, supervised methods compute the mapping both from relative positions of data (also used by non-supervised methods) and from the class labels. DR techniques [1, 2, 3] can be used as a pre-processing step for classification or clustering, or to visualize (labeled) data as a scatterplot. When mapping labeled data, there are two contradictory objectives:

- *Classification* is typical of *supervised* DR techniques: class separation is emphasized and measured with classification accuracy in the embedding space.

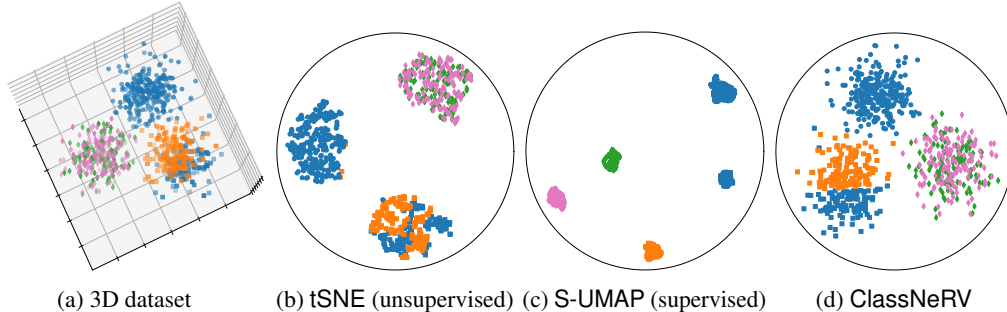

| (a) 3D dataset | (b) tSNE (unsupervised) | (c) S-UMAP (supervised) | (d) ClassNeRV |

Figure 1: ClassNeRV is designed to preserve both classes and neighbors: data are sampled from three 3D Gaussian clusters (a), a cluster of circles purely of the blue class, a cluster of squares halved by a plane separating blue and orange classes, and a cluster of diamonds with randomly distributed green and pink classes. Different planar embeddings of these data are shown: the unsupervised tSNE (b) preserves clusters well, but overlaps the orange and blue classes in the cluster of squares as it ignores the labels. The supervised S-UMAP (c) splits the clusters of squares and diamonds forcing class separation. Hence, it is misleading about the original spatial adjacency of orange and blue classes, and mixing of green and pink classes. ClassNeRV (d) is designed to better preserve the three-clusters structure as well as classes' adjacency.

- *Exploratory data analysis* is typical of *unsupervised* DR techniques which operate without knowledge of class information: data neighborhood structure is prioritized and measured as a discrepancy between data similarities in both original and embedding spaces.

These objectives derive from visual analytic tasks [3, 4]. They are contradictory unless class and data neighborhood structures match each other well in both the data and embedding spaces: each class constitutes distinct areas with no cross-class neighborhood relations. Unfortunately, this ideal case is very unlikely because the data neighborhood structure and classes do not always match in the data space, and low dimensional embeddings of high-dimensional data come with unavoidable distortions [3]: *false neighbors* which are neighboring points in the embedding but not in the data, and *missed neighbors* which are neighbors in the data but not in the embedding.

In this work, we propose ClassNeRV, a supervised DR technique to accomplish the exploratory analysis objective while taking class information into account. Our solution is similar in principle to the earlier ClassiMap [5] distance-based projection, but we derive our approach from the same well-grounded probabilistic framework as NeRV [6, 7], SNE [8], and tSNE [9]. Our solution differs notably from other previous supervised methods which tend to force class separation at the expense of neighbors' structure, *e.g.* S-Isomap [10], S-NeRV [11] and S-UMAP [12]. Figure 1 illustrates the essential characteristics of ClassNeRV.

Our contribution is two-fold: we propose **ClassNeRV** which utilizes class information to ensure a better preservation of classes when embedding high dimensional labeled data into a low dimensional space. Its stress function, derived from the unsupervised NeRV [6, 7], steers the optimization so that the unavoidable distortions of the neighborhood structure are placed where they are less harmful to the class structure. Harmful distortions are avoided by emphasizing penalization of false neighbors between classes and missed neighbors within classes. We also derive two new **class-aware quality indicators** from the standard *Trustworthiness* and *Continuity* quality indicators [13], to account specifically for the distortions affecting class preservation.

## 2   Related Work

**Unsupervised Embeddings.**   Many linear or non-linear algorithms have been previously proposed including Principal Component Analysis (PCA) [14], Self Organizing Maps (SOM) [15], isometric feature mapping (Isomap) [16], Data-Driven High Dimensional Scaling (DD-HDS) [17], Local Affine Multidimensional Projection (LAMP) [18] and Uniform Manifold Approximation and Projection (UMAP) [12]. Among the wide variety of techniques, Neighborhood Embedding (NE) techniques are efficient at preserving neighborhood structures and for computing time. Their probabilistic framework also provides a theoretical background to interpret the obtained maps in terms of a

neighbourhood retrieval task [7]. NE methods compute, for each pair of points $i, j$, the probabilistic membership of point $j$ to the neighborhood of point $i$, sometimes called similarity. These membership degrees are computed both in the data space and the embedding space. The mapping is obtained by minimizing the discrepancy of membership probabilities between these two spaces. These methods include Stochastic Neighbor Embedding (SNE) [8], t-distributed SNE (t-SNE) [9], Jensen Shannon Embedding (JSE) [19] and Neighborhood Retrieval Visualizer (NeRV) [6, 7]. SNE and t-SNE differ by the kernel used to compute their neighborhood membership degrees in the embedding space. JSE and NeRV both extend SNE to control the balance between false and missed neighbors. This tunability of NeRV and JSE makes them the best-suited for introducing supervision.

**Supervised Embeddings.** Supervised mapping methods mainly focus on *class separation*, modifying the data neighbor structures before or during the mapping. Many of them typically decrease data similarities between classes and increase them within classes, then use a standard unsupervised mapping method on the modified similarities. They may operate either on dissimilarities (distances), using metric learning, as in Supervised Locally Linear Embedding SLLE and its variants [20, 21, 22], Supervised Isomap (S-Isomap) [10, 23], Supervised NeRV (S-NeRV) [11] or Supervised UMAP S-UMAP [12], or on similarities as in the semi-supervised version of heavy-tailed tSNE [24]. Other methods rely on a global parametric mapping and optimize its parameters to *maximize class separation*, as Linear Discriminant Analysis (LDA) [25] and its kernelized variants [26, 27, 28], Neighbourhood Component Analysis (NCA) [29] and its neural-network variant [30] or Limited Rank Matrix Learning Vector Quantization (LiRaM LVQ) [31]. Class-aware tSNE (catSNE) [32] locally adapts the size of the neighbourhood to preserve based on the distribution of classes. At last, ClassiMap [5] optimizes a stress function similar to the one of Local Multidimensional Scaling (LMDS) [33], but supports the *exploratory analysis of labeled data* by taking classes into account when penalizing false and missed neighbors. However, both LMDS and ClassiMap being distance-based are sensitive to the norm concentration phenomenon in high dimensions [34] while NE techniques like NeRV mitigate this effect using *shift-invariant* membership degrees [35]. ClassNeRV derives the same principles as ClassiMap from the NeRV stress function to get the benefits of both.

## 3 ClassNeRV and Class-aware Quality Indicators

### 3.1 NeRV Stress Function

In NeRV [6, 7], the degree of membership (conditional probability) of a point $j$ to the neighborhood of another point $i$, denoted $\beta_{ij}$ in the data space and $b_{ij}$ in the embedding space, is defined as:

$$\beta_{ij} \triangleq \frac{\exp\left(-\Delta_{ij}^2/2\sigma_i^2\right)}{\sum_{k \neq i} \exp\left(-\Delta_{ik}^2/2\sigma_i^2\right)} \quad \text{and} \quad b_{ij} \triangleq \frac{\exp\left(-D_{ij}^2/2s_i^2\right)}{\sum_{k \neq i} \exp\left(-D_{ik}^2/2s_i^2\right)} \tag{1}$$

with $\Delta_{ij}$ and $D_{ij}$ the distances between points $i$ and $j$ in data and embedding spaces respectively. The distributions of membership degrees are denoted as $\beta_i \triangleq \{\beta_{ij}\}_{j \neq i}$ and $b_i \triangleq \{b_{ij}\}_{j \neq i}$. The membership degrees are shift-invariant, reducing norm concentration [35]. Scale parameters $\sigma_i$ are set to get a fixed user-chosen perplexity $p$ akin to a smooth or fuzzy measure of the number of neighbors of each point [36]: $H(\beta_i) = \log p$, with the entropy $H(\beta_i) \triangleq -\sum_{j \neq i} \beta_{ij} \log \beta_{ij}$. Here, we set embedding scale parameters $s_i$ equal to $\sigma_i$.

The NeRV stress function is a linear trade-off between two sets of Kullback–Leibler (KL) divergences:

$$\zeta_{\text{NeRV}} \triangleq \sum_i \tau \mathcal{D}_{\text{KL}}(\beta_i, b_i) + (1-\tau)\mathcal{D}_{\text{KL}}(b_i, \beta_i) \triangleq \tau \sum_{i,j \neq i} \beta_{ij} \log\left(\frac{\beta_{ij}}{b_{ij}}\right) + (1-\tau) \sum_{i,j \neq i} b_{ij} \log\left(\frac{b_{ij}}{\beta_{ij}}\right) \tag{2}$$

In Equation (2), $\tau \in [0; 1]$ controls the trade-off between $\sum_i \mathcal{D}_{\text{KL}}(\beta_i, b_i)$ penalizing *missed neighbors* (maximally when $\tau = 1$) and $\sum_i \mathcal{D}_{\text{KL}}(b_i, \beta_i)$ penalizing *false neighbors* (maximally when $\tau = 0$). When $\tau = 1$, NeRV reduces to SNE.

### 3.2 ClassNeRV Stress Function

We aim for an embedding that preserves both neighborhood structures and classes. We propose that the embedding should not *artificially separate points within the same class*, or *artificially cluster*

*together points from different classes*. Hence, we need to penalize more *within-class missed neighbors* and *between-class false neighbors* respectively. To control these class-based distortions, we further split the divergence terms in Equation (2) into *within-class* and *between-class* relations. It gives two additional distinct class-based trade-off parameters $\tau^{\in}$ and $\tau^{\notin}$ (both in $[0,1]$) and leads to the ClassNeRV stress function:

$$
\begin{aligned}
\zeta_{\text{ClassNeRV}} &\triangleq \sum_i \tau^{\in}\mathcal{D}_{\text{B}}(\beta_i^{\in}, b_i^{\in}) + (1-\tau^{\in})\mathcal{D}_{\text{B}}(b_i^{\in}, \beta_i^{\in}) + \tau^{\notin}\mathcal{D}_{\text{B}}(\beta_i^{\notin}, b_i^{\notin}) + (1-\tau^{\notin})\mathcal{D}_{\text{B}}(b_i^{\notin}, \beta_i^{\notin}) \\
&\triangleq \tau^{\in}\sum_{i,j\in\mathcal{S}_i^{\in}}\left(\beta_{ij}\log\left(\frac{\beta_{ij}}{b_{ij}}\right)+b_{ij}-\beta_{ij}\right) + (1-\tau^{\in})\sum_{i,j\in\mathcal{S}_i^{\in}}\left(b_{ij}\log\left(\frac{b_{ij}}{\beta_{ij}}\right)+\beta_{ij}-b_{ij}\right) \\
&+ \tau^{\notin}\sum_{i,j\in\mathcal{S}_i^{\notin}}\left(\beta_{ij}\log\left(\frac{\beta_{ij}}{b_{ij}}\right)+b_{ij}-\beta_{ij}\right) + (1-\tau^{\notin})\sum_{i,j\in\mathcal{S}_i^{\notin}}\left(b_{ij}\log\left(\frac{b_{ij}}{\beta_{ij}}\right)+\beta_{ij}-b_{ij}\right) \quad (3)
\end{aligned}
$$

with the sets of within-class $\mathcal{S}_i^{\in} = \{j \neq i \mid L_i = L_j\}$ and between-class $\mathcal{S}_i^{\notin} = \{j \neq i \mid L_i \neq L_j\}$ indices defined based on the class labels $L_i$ of points $i$.

Note that Bregman (B) divergences replace KL divergences of Equation (2) to ensure the positivity of the stress function. Indeed, the KL divergence is only defined for probability distributions (summing to 1), while in the four terms of $\zeta_{\text{ClassNeRV}}$, the membership degrees restricted to the sets $\mathcal{S}_i^{\in}$ or $\mathcal{S}_i^{\notin}$ sum to less than 1. Hence, KL divergence is not directly applicable to such membership degrees (otherwise it would not satisfy the known properties of a divergence, namely non-negativity and identity of indiscernibles), whereas Bregman divergence is directly applicable as is. Bregman divergence terms $\mathcal{D}_{\text{B}}(\beta_i^{\in}, b_i^{\in})$ and $\mathcal{D}_{\text{B}}(b_i^{\in}, \beta_i^{\in})$ penalize *within-class* missed and false neighbours respectively, while terms $\mathcal{D}_{\text{B}}(\beta_i^{\notin}, b_i^{\notin})$ and $\mathcal{D}_{\text{B}}(b_i^{\notin}, \beta_i^{\notin})$ penalize *between-class* missed and false neighbours respectively.

Parameters $\tau^{\in}$ and $\tau^{\notin}$ define ClassNeRV mapping behavior by weighting these terms. $\tau^{\in}$ controls the balance for penalization of false neighbors and missed neighbors *within classes*, while $\tau^{\notin}$ controls a similar balancing *between classes*. Thus, ClassNeRV **is supervised** if $\tau^{\in} > \tau^{\notin}$, *i.e.* its stress function penalizes more within-class missed neighbors, and between-class false neighbors than other distortions. The greater the $\tau^{\in} - \tau^{\notin}$ difference, the higher the level of supervision with a maximum for $\tau^{\in} = 1$ and $\tau^{\notin} = 0$. Having $\tau^{\in} < \tau^{\notin}$ however would favor missed neighbors within classes, and false neighbors between classes, encouraging same-class split and distinct-class overlap, messing up class preservation. ClassNeRV **is unsupervised** if $\tau^{\in} = \tau^{\notin}$ and then reduces to the original NeRV, the combined within and between class Bregman divergences being equal to the corresponding KL divergence, with $b_{ij} - \beta_{ij}$ terms cancelling out.

We re-parameterize $\tau^{\in}$ and $\tau^{\notin}$ as $\tau^* = (\tau^{\in} + \tau^{\notin})/2$ and $\varepsilon = (\tau^{\in} - \tau^{\notin})/2$. $\tau^* \in [0,1]$ controls the average *trade-off between penalizations of false and missed neighbors* (as $\tau$ in NeRV), while $\varepsilon \in [0, 0.5]$ controls the *level of supervision* (more supervision for greater values). The reverse transformations are: $\tau^{\in} = \tau^* + \varepsilon$ and $\tau^{\notin} = \tau^* - \varepsilon$.

Given $\tau^*$ and $\varepsilon$, the ClassNeRV embedding is obtained by minimizing the stress $\zeta_{\text{ClassNeRV}}$ in Equation (3) with respect to the coordinates of the embedded points. The optimization is performed using a multi-scale optimization approach [37] and the quasi-Newton BFGS algorithm [38]. The current implementation has a complexity of $\mathcal{O}(N^2)$ (where $N$ is the number of data points), but tree-based acceleration techniques [39, 40] could reduce it to $\mathcal{O}(N \log N)$.

### 3.3 Quality Indicators of Supervised Techniques for Exploratory Analysis

To assess the preservation of neighborhood structures, we adopt the Trustworthiness and Continuity measures [13] which are standard to evaluate unsupervised embeddings in the context of exploratory analysis [3]. These two measures separately quantify the average level of false neighbors (Trustworthiness $\mathcal{T}$) and missed neighbors (Continuity $\mathcal{C}$) for a given neighborhood size $\kappa$ as:

$$
\mathcal{T}(\kappa) \triangleq 1 - \frac{1}{\mathcal{T}_{\max}(\kappa)}\sum_{i,j\in\mathcal{F}_i(\kappa)}(\rho_{ij} - \kappa) \quad \text{and} \quad \mathcal{C}(\kappa) \triangleq 1 - \frac{1}{\mathcal{C}_{\max}(\kappa)}\sum_{i,j\in\mathcal{M}_i(\kappa)}(r_{ij} - \kappa), \quad (4)
$$

where $\rho_{ij}$ and $r_{ij}$ are the ranks of each point $j$ in the neighborhood of each point $i$ in the data and embedding spaces respectively, and $\mathcal{F}_i(\kappa)$ and $\mathcal{M}_i(\kappa)$ are the sets of indices of false and

missed neighbors of point $i$. The normalization coefficients $\mathcal{T}_{\max}(\kappa)$ and $\mathcal{C}_{\max}(\kappa)$ are defined so that $\mathcal{T}$ and $\mathcal{C}$ range from $0$ for a theoretical worst mapping to $1$ for an ideal mapping. Based on a combinatorial analysis, $\mathcal{T}_{\max}(\kappa)$ and $\mathcal{C}_{\max}(\kappa)$ are equal to either $\kappa N(2N - 3\kappa - 1)/2$ if $\kappa < N/2$ or $N(N - \kappa)(N - \kappa - 1)/2$ if $\kappa \geqslant N/2$ [41].

To assess class preservation, we also derive two new measures: Trustworthiness *restricted to between-class relations* $\mathcal{T}^{\notin}$ and Continuity *restricted to within-class relations* $\mathcal{C}^{\in}$. These *class-aware* indicators are defined as:

$$\mathcal{T}^{\notin}(\kappa) \triangleq 1 - \frac{1}{\mathcal{T}_{\max}(\kappa)} \sum_{i,j \in \mathcal{F}_i(\kappa) \cap \mathcal{S}_i^{\notin}} (\rho_{ij} - \kappa) \quad \text{and} \quad \mathcal{C}^{\in}(\kappa) \triangleq 1 - \frac{1}{\mathcal{C}_{\max}(\kappa)} \sum_{i,j \in \mathcal{M}_i(\kappa) \cap \mathcal{S}_i^{\in}} (r_{ij} - \kappa). \quad (5)$$

Notice these class-aware indicators account for only part of the distortions considered by their unsupervised counterparts, so that $\mathcal{T}^{\notin} \geqslant \mathcal{T}$ and $\mathcal{C}^{\in} \geqslant \mathcal{C}$. Thus, they may reach $1$ as long as any remaining distortions do not affect between-class Trustworthiness and within-class Continuity.

Another standard quality indicator in supervised dimensionality reduction is the accuracy of a leave-one-out $k$-Nearest neighbors ($k$-NN) classifier in the embedding space [3]. This indicator makes sense when the embedding is used to perform classification, but it focuses on classes only and fails to account for neighborhood structure preservation as shown in Figure 3. Supplementary material presents extensive results for the $k$-NN gain [32] derived from that indicator.

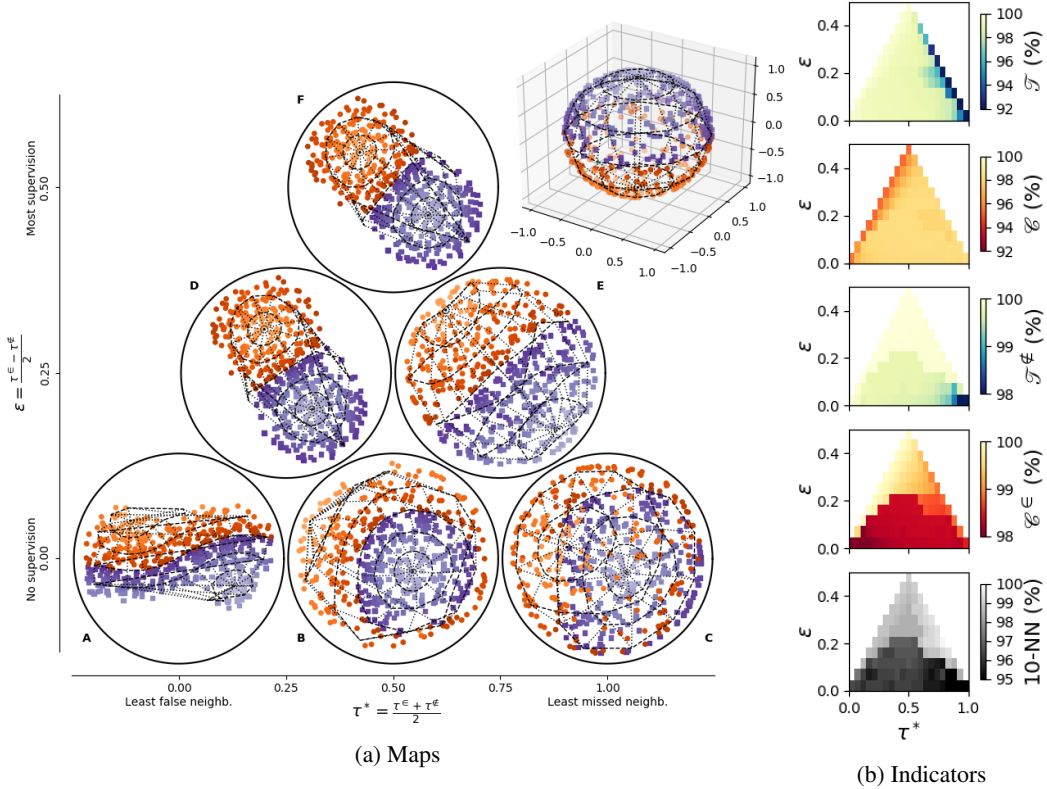

(a) Maps

(b) Indicators

Figure 2: Sensitivity of ClassNeRV to the trade-off parameters while embedding the $512$ data of the 3D *Globe* dataset (Section 4.2) in the $\{\tau^*, \varepsilon\}$-parameter space. The left panel (a) presents five 2-dimensional embeddings with parallels (dashed lines) and meridians (dotted lines) positioned *a posteriori*, as well as a 3D representation of the dataset (top right). The right panel (b) shows quantitative evaluation with indicators defined in Section 3.3, with from top to bottom Trustworthiness $\mathcal{T}$, Continuity $\mathcal{C}$, between-class Trustworthiness $\mathcal{T}^{\notin}$ and within-class Continuity $\mathcal{C}^{\in}$ for a given neighborhood size of $\kappa = 32$, and $10-$NN classification accuracy ($99\%$ for the original 3D data). Each pixel of the heat map corresponds to one embedding of the globe. $\tau^*$ controls the overall trade-off between false neighbors (most likely with $\tau^* = 1$) and missed neighbors (most likely with $\tau^* = 0$), while $\varepsilon$ controls the level of supervision.

# 4 Experiments

## 4.1 Objectives, Data, Techniques

We illustrate the main characteristics of ClassNeRV compared to other unsupervised and supervised DR techniques, on a 3D toy dataset (*Globe*) and on two real high dimensional datasets (*Isolet 5 and Digits*). The *Globe* dataset (Section 4.2) contains $512$ data randomly distributed on the surface of the unit sphere in the three-dimensional Euclidean space $\mathbb{R}^3$ (Figure 2a). The two classes (blue and red points) correspond to the two hemispheres divided at the equator. These data cannot be embedded in the plane without distortions, so the final map depends on the trade-off set between the neighborhood ($\tau^*$) and class ($\varepsilon$) penalizations (See Section 3.2). The *Isolet 5* dataset [42, 43] (Section 4.3) contains $1\,559$ recordings of English pronunciation of spoken letters evenly distributed in 26 classes, described by $617$ features. The *Digits* dataset [44, 43] (Section 4.4) contains $3\,823$ images of handwritten digits. The $8 \times 8$ pixels images (64D data points) are separated in 10 classes (one per digit). True class labels as well as randomly generated labels are considered to evaluate the robustness to mislabeling. A random subset of $500$ samples is considered to ease the readability of the maps in Figure 6.

We compare ClassNeRV to unsupervised PCA [14], Isomap [16], UMAP [12], tSNE [9], and NeRV [6, 7], and to supervised NCA [29], S-Isomap [10], ClassiMap [5] and S-UMAP [12]. Implementations of PCA, Isomap, NCA, tSNE, UMAP and S-UMAP are from scikit-learn (version 0.22.1) [45] and umap-learn (version 0.3.10) [46] Python libraries. S-Isomap, ClassiMap, NeRV and ClassNeRV use our own implementations [47]. All off-the-shelf algorithms are set with default parameters, except for tSNE initialization, for which we used PCA instead of random (so that all methods benefit from a spectral initialization). NeRV and ClassNeRV use the optimization described in Section 3.2. The final perplexity in our multi-scale optimization is set to $p = 32$ for *Globe* and $p = 30$ for *Isolet*, for comparability with tSNE default $p = 30$. We use the Euclidean distance for all techniques and datasets. Qualitative results are given as scatterplot representations of the embeddings with circular frame following standard guidelines [3, 48]. Quantitative results are computed with unsupervised and class-aware indicators described in Section 3.3. Additional runs of stochastic methods are shown in the supplementary material showing similar results.

## 4.2 Two Hemispheres Globe Example

Figure 2a shows ClassNeRV embeddings of the *Globe* data in the $(\tau^*, \varepsilon)$-parameter space. Maps A, B and C correspond to the original NeRV, *i.e.* without any supervision ($\varepsilon = 0$, $\tau^\in = \tau^\notin$). On map A, false neighbors are the most penalized ($\tau^\in = \tau^\notin = 0$), allowing some missed neighbors, so that the sphere is torn along a meridian and unfolded. On map C, missed neighbors are the most penalized ($\tau^\in = \tau^\notin = 1$), corresponding to the original SNE mapping [8], letting false neighbors, so that the sphere is squashed, blending the red and blue classes. Map B corresponds to NeRV with a balanced mix of missed and false neighbors ($\tau^\in = \tau^\notin = 0.5$). Adding some supervision ($\varepsilon = 0.25$), map D penalizes moderately within-class missed neighbors ($\tau^\in = 0.5$) and strongly all-class false neighbors ($\tau^* = 0.25$), encouraging less torn classes than in map A. Conversely, map E penalizes moderately between-class false neighbors ($\tau^\notin = 0.5$) and strongly all-class misses ($\tau^* = 0.75$), encouraging more class separation than in map C, but more within-class false neighbors than in map D. Finally, map F corresponds to the maximum level of supervision ($\varepsilon = 0.5$ with $\tau^\in = 1$ and $\tau^\notin = 0$), penalizing the most between-class false neighbors (the least class overlap) and within-class missed neighbors (the most class cohesion). To get a detailed idea of the individual impact of each of ClassNeRV stress sub-terms (Equation (3)), an ablation study is available in supplementary material.

Figure 2b quantitatively supports these qualitative observations. Top heat maps show *structure preservation* indicators in the same $\{\tau^*, \varepsilon\}$-parameter space: when $\tau^*$ increases, the amount of false neighbors ($\mathcal{T}$ in blue) increases, while the amount of missed neighbors ($\mathcal{C}$ in red) decreases. Bottom heat maps show *class preservation* indicators in that space: when the level of supervision $\varepsilon$ increases, there is less class overlap ($\mathcal{T}^\notin$ in blue) and less class split ($\mathcal{C}^\in$ in red). The classification heat map also shows a greater class accuracy ($10-$NN in grey) coherent with the higher class cohesion and lower class overlap encouraged by greater $\varepsilon$.

Figures 3 and 4 show qualitative and quantitative comparison of the most supervised version of ClassNeRV ($\varepsilon = 0.5$) to other techniques. A qualitative analysis of supervised embeddings shows that ClassNeRV (Figure 2a, maps D, E and F), ClassiMap (Figure 3c) and NCA (Figure 3b) maps

all preserve the actual 3D linear separability and adjacency of the two classes. However, S-Isomap (Figure 3a) generates a false class overlap, while S-UMAP (Figure 3d) wrongly splits classes apart.

Figure 4 (top row), shows that ClassNeRV (—) reaches similar levels of structure preservation as tSNE (—), or UMAP (—) unsupervised techniques (I), with higher trustworthiness in general (less false neighbors). It also preserves classes far better (II), as expected against unsupervised techniques. Regarding supervised techniques, ClassNeRV (—) obtains both better structure (III) and class (IV) preservation than NCA (—), ClassiMap (—) and S-Isomap (—). For S-UMAP (—), the class-independent indicators (III) are similar to ClassNeRV, except for larger neighborhood size $\kappa$, due to S-UMAP's tendency to over-separate classes (Figure 3d) actually adjacent in the *Globe* data.

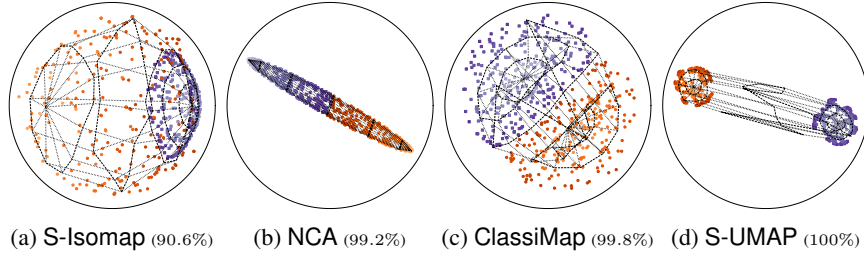

(a) S-Isomap (90.6%)    (b) NCA (99.2%)    (c) ClassiMap (99.8%)    (d) S-UMAP (100%)

Figure 3: Embeddings of the *Globe* dataset obtained with state-of-the-art dimensionality reduction techniques, to be compared with the most supervised ClassNeRV ($\varepsilon = 0.5$) in map F of Figure 2a. 10-NN accuracy is given for each technique. Analysis in Section 4.2.

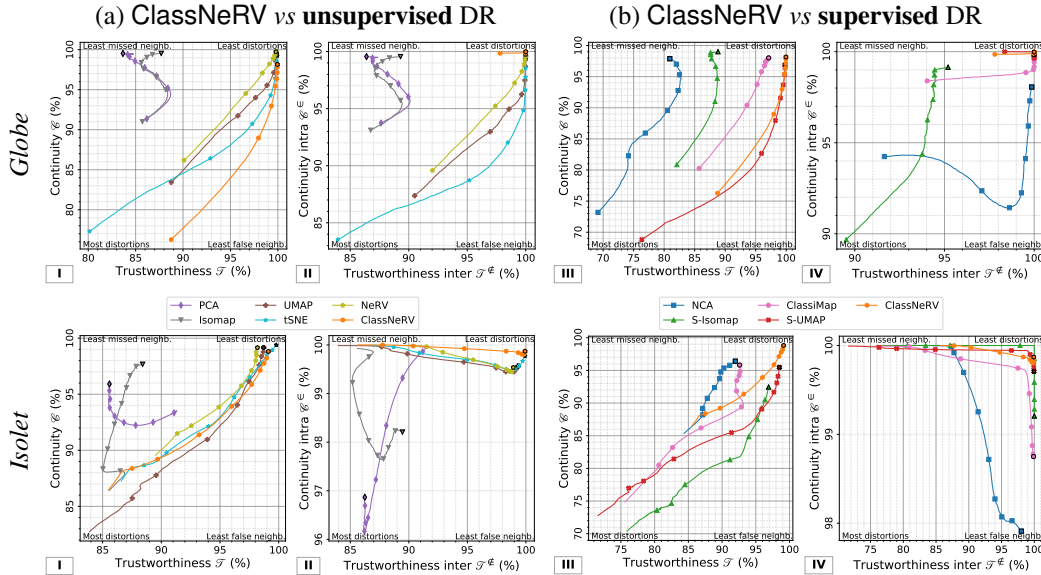

Figure 4: Quality indicators (Section 3.3) for ClassNeRV with maximum supervision ($\varepsilon = 0.5$) and state-of-the-art embeddings of *Globe* (top) and *Isolet* (bottom) datasets. Each polyline corresponds to the series of indicator values computed with $\kappa$ from 2 to $\lfloor N/2 \rfloor$ (with markers indicating $\kappa \in \{2, 4, 8, 16 \dots \}$ and $\kappa = 2$ highlighted with black outline). Unsupervised (I, II) and supervised (III, IV) DR techniques (color coded) are compared to ClassNeRV (orange) in the space of standard structure preservation indices $(\mathcal{T}, \mathcal{C})$ (I, III), and in the space of class preservation indices $(\mathcal{T}^{\notin}, \mathcal{C}^{\in})$ (II, IV). Perfect embeddings at top right corner. Analysis in sections 4.2 (*Globe*) and 4.3 (*Isolet*).

## 4.3 Isolet Dataset

The 10-NN confusion matrix computed in the 617D data space (Figure 5a) shows all classes with less than 90% accuracy or confused with at least 10% of another class (The full confusion matrix is given in the supplementary material). Several classes partially overlap likely due to similar sounds

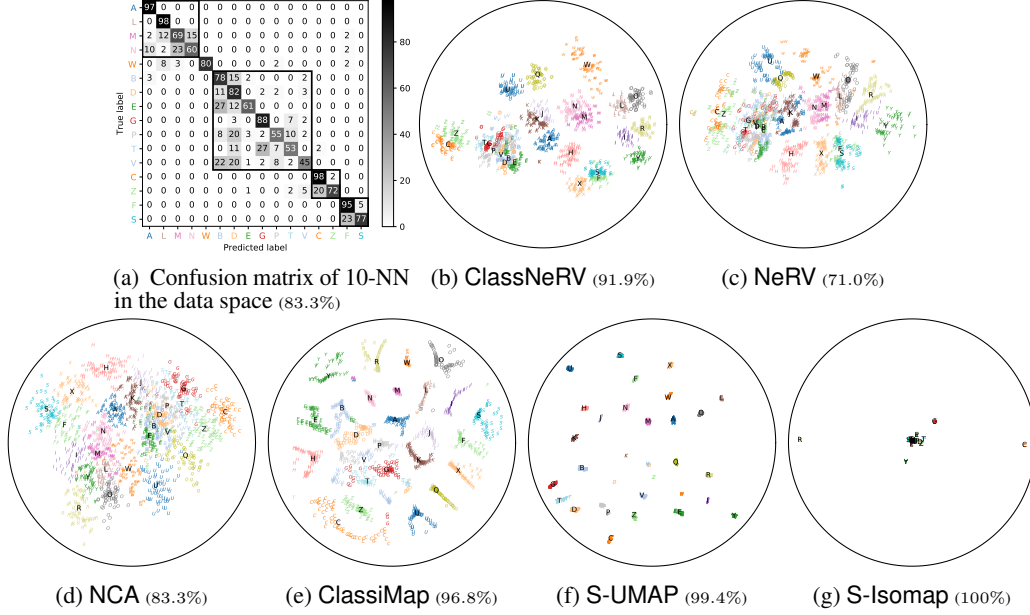

(a) Confusion matrix of 10-NN in the data space (83.3%)  (b) ClassNeRV (91.9%)  (c) NeRV (71.0%)

(d) NCA (83.3%)  (e) ClassiMap (96.8%)  (f) S-UMAP (99.4%)  (g) S-Isomap (100%)

Figure 5: Confusion matrix (a) of a 10-NN classifier for the *Isolet 5* dataset showing several classes overlap in the data space. Only the most confused classes are shown in the matrix. Supervised embeddings should preserve this actual class structure. The most supervised ClassNeRV ($\varepsilon = 0.5$) (b) is compared to NeRV (c) and supervised methods (d,e,f,g). Analysis in Section 4.3. 10-NN leave-one-out accuracy score is given for the original data (a) and all the embeddings (b to g).

indicated by phonetic symbols, *e.g.* letters F [ɛf] and S [ɛs] (group FS), letters C [siː] and Z [ziː] (group CZ), letters A [eɪ], L [ɛl], M [ɛm] and N [ɛn] (group ALMN), or letters B [biː], D [diː], E [iː], G [dʒiː], P [piː], T [tiː] and V [viː] (group BDEGPTV).

Figure 5 shows the embeddings for qualitative comparison. NCA (Figure 5d) tends to artificially overlap classes, for instance QV and JP. Conversely, S-UMAP and S-Isomap tend to over-separate classes, completely ignoring the actual overlapping of letters in groups FS, CZ, ALMN, BDEGPTV shown by the confusion matrix. ClassiMap (Figure 5e) better preserves the BDEGPTV group but fails to represent others like MN or CZ. NeRV manages to preserve groups despite being unsupervised, showing that classes of these groups actually overlap in the data space. Yet, it may artificially increase that overlap as observed in Figure 1b. Lastly, ClassNeRV (Figure 5b) provides a more trustworthy representation of the classes, better showing the stronger confusion of letters within MN, CZ, FS and BDEGPTV groups by putting them adjacent in the map, while keeping other letters separated as they actually are in the data space. Hence, the ClassNeRV map is more trustworthy to help a domain expert discover that, in this feature space, letters are strongly grouped by vowel sounds (BDEGPTV, FS or MN), with a secondary effect of the consonant.

The 10-NN classification scores (Figure 5) increase with class separation observed on these maps. Yet, as S-Isomap reaches the maximal value 100% by artificially separating all classes irrespective of the actual classes' confusion in the data space, this classification indicator cannot assess the class structure preservation. Thus, it cannot be used to evaluate if an embedding is trustworthy enough to support the *exploratory analysis of labeled data*.

The quantitative comparison of ClassNeRV (—) with unsupervised techniques on Figure 4a (bottom row) shows it preserves the neighborhood structure as well as NeRV (—) and UMAP (—), but slightly worse than tSNE (—) (I). It preserves classes better than all unsupervised techniques as expected especially for small $\kappa$ (II). Regarding supervised techniques, Figure 4b (bottom row) demonstrates that ClassNeRV (—) preserves the neighborhood structure of *Isolet data* far better than the other supervised methods (III). NCA (—) generates more false neighbors ($\mathcal{T} < \mathcal{C}$), while S-Isomap (—) and S-UMAP (—) generate more missed neighbors, consistently with the embeddings (Figure 5). All techniques except NCA (—) achieve similar good results for class preservation (IV), with an advantage at small and high scales $\kappa$ for ClassNeRV.

## 4.4 Handwritten digits

Figure 6 presents maps reached by NeRV and ClassNeRV applied to the *digits* dataset with true and random labels. The case of random labels allows to study the behaviour of supervised methods when the structure and classes are decorrelated. Separating these random classes would distort the neighborhoods, which may be seen as an over-separation. We can observe on Figures 6a and 6b that ClassNeRV relying on true class-information reaches a better preservation of classes than NeRV for the *digits* dataset. Conversely, ClassNeRV with random labels (Figure 6d) does not lead to obvious over-separation of the random classes. Those observations are confirmed by Figure 7. In addition, ClassNeRV shows better preservation of both structure and true classes than other supervised methods, when fed with the true labels and even more with the random labels (Figure 7).

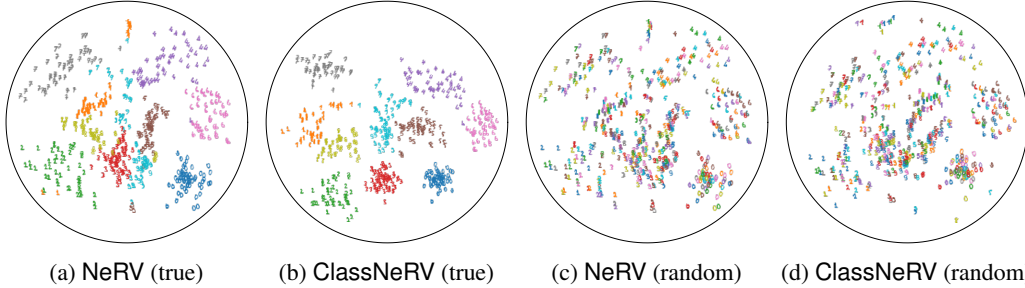

(a) NeRV (true)          (b) ClassNeRV (true)          (c) NeRV (random)          (d) ClassNeRV (random)

Figure 6: NeRV and ClassNeRV maps for the *digits* dataset, points being represented by their associated image coloured based on the class supplied to the algorithm (either true or random). Given that NeRV is not supervised, maps (a) and (c) have strictly identical points position. Conversely, differences may be observed between ClassNeRV maps (b) and (d). (b) shows better class-preservation than (a) due to the valid class-information, while no obvious over-classification may be noted on (d) with completely unreliable class-information (random labelling). Analysis in Section 4.4.

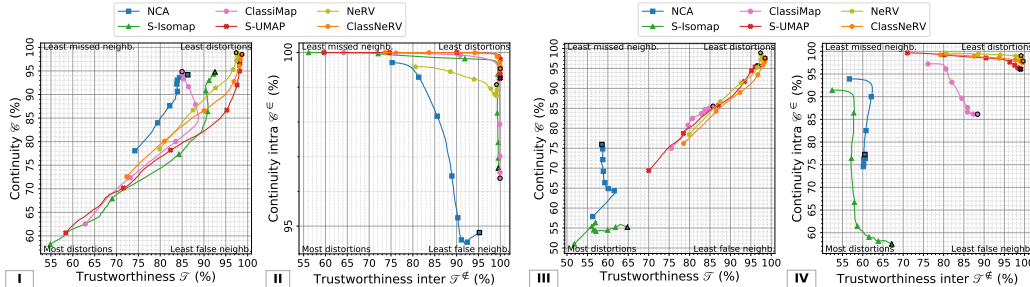

Figure 7: Quality indicators for ClassNeRV, NeRV and supervised methods on the *digits* dataset with true labels (I, II) and random labels (III, IV). Class-aware indicators $(\mathcal{T}^{\notin}, \mathcal{C}^{\in})$ (II, IV) quantify the preservation of the true classes. Note that NeRV does not account for the classes, hence its results in (III, IV) are identical to those in (I, II). Analysis in Section 4.4.

## 5 Conclusion

ClassNeRV allows data scientists to control class and structure preservation in low dimensional embeddings for exploratory data analysis of labeled data. Such analysis can for instance help detect whether classes are well-separated in a given feature space, which may lead to question the labels (labelling errors) or the features (feature engineering). Our experiments showed that supervised DR techniques tend to over-separate classes that are actually adjacent or overlapping in the data space, while unsupervised techniques totally ignore classes. In contrast, the fully supervised setting of ClassNeRV better preserves actual class structure than supervised DR techniques, while preserving neighborhood structure as well as state-of-the-art unsupervised DR techniques. Future work will extend this approach to other neighborhood embedding techniques, such as tSNE [9] or JSE [19], and consider the semi-supervised framework.

## 6 Broader impact

This work proposes improvement on a dimensionality reduction technique for exploratory data analysis. Dimensionality reduction is intended to support data scientists in analyzing multidimensional data, and can also be used to visualize high-dimensional data representing physical objects or persons in a 2D map for the lay public to get an overview of the main groups of objects/persons based on similarities in their corresponding data. It is agnostic to the nature of the objects/persons represented by the data. Better understanding of trends and variation in large-scale datasets can improve the ability of society to learn about important phenomena. However, dimensionality reduction can generate biased representation of these objects/persons, either due to the inherent bias of the data themselves (over or under represented classes of objects/persons, missing or irrelevant features artificially gathering or separating (classes of) objects/persons), or due to unavoidable projection biases, called distortions, [3] artificially gathering in the 2D representation actually separated (classes of) objects/persons, or artificially separating in the 2D representation actually similar (classes of) objects/persons. The proposed ClassNeRV method is exactly intended to reduce this second type of bias.

## 7 Funding disclosure

The work of Denys Dutykh has been supported by the French National Research Agency, through Investments for Future Program (ref. $ANR-18-EURE-0016$ — Solar Academy). Jaakko Peltonen was supported by Academy of Finland projects 313748 and 327352.

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
