[Supplementary Material]

# Steering Distortions to Preserve Classes and Neighbors in Supervised Dimensionality Reduction — Supplementary Material —

**Benoît Colange**
Univ. Grenoble Alpes, INES,
F-73375, Le Bourget du Lac, France
benoit.colange@cea.fr

**Jaakko Peltonen**
Tampere University, Faculty of Information Technology
and Communication Sciences, Finland
jaakko.peltonen@tuni.fi

**Michaël Aupetit**
Qatar Computing Research Institute,
Hamad bin Khalifa University, Doha, Qatar
maupetit@hbku.edu.qa

**Denys Dutykh**
Univ. Grenoble Alpes, Univ. Savoie Mont Blanc,
CNRS, LAMA, 73000 Chambéry, France
denys.dutykh@univ-smb.fr

**Sylvain Lespinats**
Univ. Grenoble Alpes, INES,
F-73375, Le Bourget du Lac France
sylvain.lespinats@cea.fr

## Abstract

This supplementary material demonstrates the **link between NeRV and Class-NeRV** in its unsupervised case (Section 1), illustrates individual effect of each sub-term of ClassNeRV stress with an ablation study (Section 2) and presents the **full confusion matrix** of the 10-NN classifier on the *Isolet* dataset (Section 3). It also provides the **parameters of the DR techniques** (Section 4), **additional experiments** to support our claim while accounting for randomness of stochastic methods (Section 5), quantitative comparison with unsupervised DR methods for the digits dataset (Section 6) and **additional supervised indicators** (Section 7).

## 1 Equivalence of unsupervised ClassNeRV and NeRV

ClassNeRV with parameters $\tau^{\in} = \tau^{\notin} = \tau^*$ (*i.e.* $\varepsilon = 0$) where $\tau^* \in [0,1]$, is unsupervised and corresponds to NeRV with trade-off parameter $\tau^*$. Indeed, ClassNeRV stress is given by (Equation 3):

$$\zeta_{\text{ClassNeRV}} \triangleq \tau^{\in} \sum_{i,j \in \mathcal{S}_i^{\in}} \left( \beta_{ij} \log \left( \frac{\beta_{ij}}{b_{ij}} \right) + b_{ij} - \beta_{ij} \right) + (1 - \tau^{\in}) \sum_{i,j \in \mathcal{S}_i^{\in}} \left( b_{ij} \log \left( \frac{b_{ij}}{\beta_{ij}} \right) + \beta_{ij} - b_{ij} \right)$$

$$+ \tau^{\notin} \sum_{i,j \in \mathcal{S}_i^{\notin}} \left( \beta_{ij} \log \left( \frac{\beta_{ij}}{b_{ij}} \right) + b_{ij} - \beta_{ij} \right) + (1 - \tau^{\notin}) \sum_{i,j \in \mathcal{S}_i^{\notin}} \left( b_{ij} \log \left( \frac{b_{ij}}{\beta_{ij}} \right) + \beta_{ij} - b_{ij} \right).$$

Hence, for $\tau^{\in} = \tau^{\notin} = \tau^*$, it may be factored by $\tau^*$ and $(1 - \tau^*)$, so that the sums of within class and between class terms collapse, to form sums over all pairs. This leads to:

$$\zeta_{\text{ClassNeRV}} = \tau^* \sum_{i,j \neq i} \left( \beta_{ij} \log \left( \frac{\beta_{ij}}{b_{ij}} \right) + b_{ij} - \beta_{ij} \right) + (1 - \tau^*) \sum_{i,j \neq i} \left( b_{ij} \log \left( \frac{b_{ij}}{\beta_{ij}} \right) + \beta_{ij} - b_{ij} \right).$$

Knowing that $\sum_{j\neq i}\beta_{ij} = \sum_{j\neq i} b_{ij} = 1$ (due to the normalization in Equation 1), $\sum_{j\neq i}\beta_{ij}$ and $\sum_{j\neq i} b_{ij}$ cancel each other out, so that $b_{ij} - \beta_{ij}$ and $\beta_{ij} - b_{ij}$ terms may be removed from the above equation, giving:

$$\zeta_{\text{ClassNeRV}} = \tau^* \sum_{i,j\neq i} \beta_{ij} \log\left(\frac{\beta_{ij}}{b_{ij}}\right) + (1-\tau^*) \sum_{i,j\neq i} b_{ij} \log\left(\frac{b_{ij}}{\beta_{ij}}\right).$$

As a result, the Bregman divergence becomes a Kullback-Leibler divergence and ClassNeRV stress equals the stress of NeRV (Equation 2), with trade-off $\tau^*$.

## 2 Ablation study

To get a finer assessment of the individual impacts of each sub-term of ClassNeRV stress (Equation 3), we perform an ablation study. We rewrite the ClassNeRV stress with weights $w = (w_1, w_2, w_3, w_4)$:

$$\zeta_{\text{ClassNeRV}} \triangleq \frac{1}{2} \sum_i w_1 \cdot \mathcal{D}_{\text{B}}(\beta_i^{\in}, b_i^{\in}) + w_2 \cdot \mathcal{D}_{\text{B}}(b_i^{\in}, \beta_i^{\in}) + w_3 \cdot \mathcal{D}_{\text{B}}(\beta_i^{\notin}, b_i^{\notin}) + w_4 \cdot \mathcal{D}_{\text{B}}(b_i^{\notin}, \beta_i^{\notin})$$

The case of $w = (1, 1, 1, 1)$ being equivalent to ClassNeRV with $\tau^{\in} = \tau^{\notin} = 0.5$. The ablation study consists in successively removing each term of the ClassNeRV stress, zeroing the weights $w_1$, $w_2$, $w_3$ and $w_4$ one at a time, the others being equal to 1.

(a) Removing $\mathcal{D}_{\text{B}}(\beta_i^{\in}, b_i^{\in})$ $w = (0, 1, 1, 1)$    (b) Removing $\mathcal{D}_{\text{B}}(b_i^{\in}, \beta_i^{\in})$ $w = (1, 0, 1, 1)$    (c) Removing $\mathcal{D}_{\text{B}}(\beta_i^{\notin}, b_i^{\notin})$ $w = (1, 1, 0, 1)$    (d) Removing $\mathcal{D}_{\text{B}}(b_i^{\notin}, \beta_i^{\notin})$ $w = (1, 1, 1, 0)$

Figure 1: Ablation study for the *globe* dataset.

(a) Removing $\mathcal{D}_{\text{B}}(\beta_i^{\in}, b_i^{\in})$ $w = (0, 1, 1, 1)$    (b) Removing $\mathcal{D}_{\text{B}}(b_i^{\in}, \beta_i^{\in})$ $w = (1, 0, 1, 1)$    (c) Removing $\mathcal{D}_{\text{B}}(\beta_i^{\notin}, b_i^{\notin})$ $w = (1, 1, 0, 1)$    (d) Removing $\mathcal{D}_{\text{B}}(b_i^{\notin}, \beta_i^{\notin})$ $w = (1, 1, 1, 0)$

Figure 2: Ablation study for the *digits* dataset.

When removing the term $\mathcal{D}_{\text{B}}(\beta_i^{\in}, b_i^{\in})$ which penalizes within-class missed neighbours ($w = (0, 1, 1, 1)$), the orange class of the *globe* is torn (Figure 1a).

The removal of $\mathcal{D}_{\text{B}}(b_i^{\in}, \beta_i^{\in})$ penalizing within-class false neighbours ($w = (1, 0, 1, 1)$), induces the collapse of the two-hemispheres (Figure 1b).

Ablation of $\mathcal{D}_{\text{B}}(\beta_i^{\notin}, b_i^{\notin})$ which prevents between-class missed neighbours ($w = (1, 1, 0, 1)$), leads to separating the two hemisphere along the the equator (Figure 1c).

Removal of $\mathcal{D}_B(b_i^{\notin}, \beta_i^{\notin})$, that avoid between-class false neighbours ($w = (1,1,1,0)$) leads to the collapse of the two hemisphere onto each other (Figure 1d).

Figure 2 shows the equivalent study for the digits dataset with true labels. The effect is especially clear for Figure 2d, for which several classes strongly overlap.

## 3 Confusion matrix

We present here the full confusion matrix of a leave one out 10-NN classifier on the *Isolet 5* dataset, for which several classes have been filtered out in the paper. Classes are reordered so as to get a near block diagonal structure. We may notice some light confusion between letters J and K, and between letters Q and U, that was not visible on the filtered confusion matrix. Conversely, we may see some classes involved in very few or no cases of confusion, with for example letter Y involved in $0\%$ of confusion (considering both its associated row and column).

The leave one-out-classifier $k$-NN classifier, also used for $k$-NN accuracy, attributes to each point $i$ the majority label (winner takes all strategy) of its $k$-nearest neighbours (among all points except $i$) and the equality case is decided randomly.

Figure 3: Full confusion matrix for the Isolet dataset

## 4 Technique parameters

Table 1: Default parameterization and implementations library of the DR techniques used in all experiments of the main paper.

| Method | Initialization | Neighbourhood size | Implementation |
|---|---|---|---|
| PCA | N.A. | N.A. | scikit-learn (0.22.1) |
| Isomap | N.A. | 5 | scikit-learn (0.22.1) |
| UMAP | Spectral | 15 | umap-learn (version 0.3.10) |
| tSNE | PCA/random | 30 | scikit-learn (0.22.1) |
| NeRV | PCA | 32/30 | author implementation |
| NCA | LDA | N.A. | scikit-learn (0.22.1) |
| S-Isomap | N.A. | 5 | author implementation |
| Classimap | DD-HDS | 5 | author implementation |
| S-UMAP | Spectral | 15 | umap-learn (version 0.3.10) |
| ClassNeRV | PCA | 32/30 | author implementation |

## 5 Evaluation indicators with varying random states

Some of the dimensionality reduction techniques to which ClassNeRV is compared are stochastic, so their results may vary due to randomization in the optimization of the stress function (*e.g.* stochastic gradient methods), or in the initialization of the embedding. The former case happens for tSNE, UMAP, S-UMAP and Classimap. The latter case happens for tSNE, for which the initialization may be either PCA or random. Conversely, PCA, Isomap and S-Isomap are designed to systematically find the global minimum of their stress functions by Singular Value Decomposition, while the implementations used for NCA, NeRV and ClassNeRV use deterministic initialization and optimization methods.

To assess the robustness of the quantitative analysis in section 4 of the main paper, we present the minimum, median and maximum value of the indicators over 30 runs (random initialization) of each technique. Here, indicators are presented individually (*e.g* $\mathcal{T}$ vs $\kappa$) rather than by pairs (*e.g.* $\mathcal{T}$ vs $\mathcal{C}$). This less compact view gives more space to see the variations for each neighborhood size $\kappa$. It also eases the comparison of a pair of methods for a given indicator and a given $\kappa$ scale.

The variability shown by Figures 4, 5, 6 and 7 does not hinder the repeatability of the results already observed in the main paper.

Table 2: ClassNeRV *vs* State-of-the-Art techniques (SofA) on *Globe* and *Isolet* datasets

|  | Unsupervised DR | Supervised DR |
|---|---|---|
| Neighbors preservation | ClassNeRV $\approx$ best SofA (Fig. 4) | ClassNeRV $>$ SofA (Fig. 6) |
| Classes preservation | ClassNeRV $>$ SofA (Fig. 5) | ClassNeRV $\approx$ best SofA (Fig. 7) |

These supplemental results on the Globe and Isolet datasets support our main claim (see Table 2):

- ClassNeRV is as good as the best unsupervised DR for neighbors preservation (Fig. 4);
- ClassNeRV dominates unsupervised DR for classes preservation indicators (Fig. 5);
- ClassNeRV dominates supervised DR for neighbors preservation indicators (Fig. 6);
- ClassNeRV is as good as the best supervised DR for classes preservation (Fig. 7).

## 5.1 ClassNeRV is as good as best Unsupervised DR for Neighbors Preservation

(a) Trustworthiness for the *Globe* dataset

(b) Continuity for the *Globe* dataset

(c) Trustworthiness for the *Isolet* dataset

(d) Continuity for the *Isolet* dataset

Figure 4: ClassNeRV is as good as the best unsupervised DR techniques regarding neighbors preservation on both *Globe* (top) and *Isolet* (bottom). Quality indicators (higher values indicating better mapping) are plotted against their scale parameter $\kappa$ (from to to $N/2$ with $N$ the number of data points). The neighbourhood size parameters of UMAP (15), tSNE (30) and NeRV/ClassNeRV (32 for *Globe* and 30 for *Isolet*) are marked by vertical dashed lines and give the scale at which optimal neighbor preservation is expected. Shaded areas present the range of variation between the minimum and maximum value of indicators for the 30 runs of a method, while black lines show the median values.

## 5.2 ClassNeRV dominates Unsupervised DR for Class Preservation

(a) Between-class Trustworthiness for the *Globe* dataset

(b) Within-class Continuity for the *Globe* dataset

(c) Between-class Trustworthiness for the *Isolet* dataset

(d) Within-class Continuity for the *Isolet* dataset

Figure 5: ClassNeRV dominates unsupervised DR techniques regarding classes preservation on both *Globe* (top) and *Isolet* (bottom). Quality indicators (higher values indicating better mapping) are plotted against their scale parameter $\kappa$ (from to to $N/2$ with $N$ the number of data points). The neighbourhood size parameters of UMAP (15), tSNE (30) and NeRV/ClassNeRV (32 for *Globe* and 30 for *Isolet*) are marked by vertical dashed lines and give the scale at which optimal neighbor preservation is expected. Shaded areas present the range of variation between the minimum and maximum value of indicators for the 30 runs of a method, while black lines show the median values.

## 5.3    ClassNeRV dominates Supervised DR for Neighbors Preservation

(a) Trustworthiness for the *Globe* dataset

(b) Continuity for the *Globe* dataset

(c) Trustworthiness for the *Isolet* dataset

(d) Continuity for the *Isolet* dataset

Figure 6: ClassNeRV dominates supervised DR techniques regarding neighbors preservation on both *Globe* (top) and *Isolet* (bottom). Quality indicators (higher values indicating better mapping) are plotted against their scale parameter $\kappa$ (from to to $N/2$ with $N$ the number of data points). The neighbourhood size parameters of S-UMAP (15) and ClassNeRV (32 for *Globe* and 30 for *Isolet*) are marked by vertical dashed lines and give the scale at which optimal neighbor preservation is expected. Shaded areas present the range of variation between the minimum and maximum value of indicators for the 30 runs of a method, while black lines show the median values.

## 5.4 ClassNeRV is as good as best Supervised DR for Class Preservation

(a) Between-class Trustworthiness for the *Globe* dataset

(b) Within-class Continuity for the *Globe* dataset

(c) Between-class Trustworthiness for the *Isolet* dataset

(d) Within-class Continuity for the *Isolet* dataset

Figure 7: ClassNeRV is as good as the best supervised DR techniques regarding classes preservation on both *Globe* (top) and *Isolet* (bottom). Quality indicators (higher values indicating better mapping) are plotted against their scale parameter $\kappa$ (from to to $N/2$ with $N$ the number of data points). The neighbourhood size parameters of S-UMAP (15) and ClassNeRV (32 for *Globe* and 30 for *Isolet*) are marked by vertical dashed lines and give the scale at which optimal neighbor preservation is expected. Shaded areas present the range of variation between the minimum and maximum value of indicators for the 30 runs of a method, while black lines show the median values.

# 6  ClassNeRV vs unsupervised methods for the digits dataset

The results of ClassNeRV against unsupervised methods for the *digits* are coherent with those obtained with other datasets. The structure preservation is comparable, with some additional distortions at high scales $\kappa$, while the class-preservation is improved over unsupervised methods.

Figure 8: ClassNeRV compared with unsupervised techniques for the *digits* dataset

# 7  Additional supervised indicator

The $k$-NN gain [1] is defined for a number of neighbours $k$ as:

$$G(k) = \frac{1}{N} \sum_i \frac{|n_i(k) \cap \mathcal{S}_i^\in|}{|n_i(k)|} - \frac{|\nu_i(k) \cap \mathcal{S}_i^\in|}{|\nu_i(k)|}, \tag{1}$$

where $n_i(k)$ and $\nu_i(k)$ are respectively the sets of the $k$-nearest neighbours of $i$ in the embedding and data space, and $|\cdot|$ denotes the cardinal of a set. Hence, it is a measure of the average difference between the embedding and data space of the proportion of $k$-nearest neighbours of each point $i$ that are within-class neighbours. A positive value suggests that the performances of a $k$-NN classifier should be higher in the map than in the data space. As such, the $k$-NN gain is an indicator based on classification accuracy taking into account the effective separation of classes in the data space.

Figure 9 presents that score for several values of $k$ for several datasets. ClassNeRV leads to higher values than unsupervised DR for that indicator, except for a few values of $k$ in the case of digits with random labels. Note that in that case the range of value is very small (from $-2\%$ to $+5\%$).

Compared with supervised DR, ClassNeRV has lower gain than all methods excepts Classimap and S-Isomap for the *globe* dataset, and NCA for *Isolet* and *digits* with true labels. Yet, for the specific case of *digits* with random labels, where the classes are unrelated to the structure, the gain of ClassNeRV is close to $0$, as opposed to other supervised methods. This confirms that ClassNeRV does not over-separate the random classes, contrary to other supervised methods.

# References

[1] C. de Bodt, D. Mulders, D. L. Sánchez, M. Verleysen, and J. A. Lee, "Class-aware t-SNE: cat-SNE.," in *ESANN*, 2019.

Figure 9: $k$-NN gain for all possible values of $k$ for the 4 datasets and all unsupervised and supervised DR techniques.