[Reviews · NeurIPS 2020]

Review 1

Summary and Contributions: This paper presents a new supervised dimensionality reduction approach that is based on distinguishing between two types of distortions introduced by mapping high dimensional data to a low dimensional representation. Namely, these are the distortion to local neighborhoods, encoding the unsupervised intrinsic geometry of the data, and the distortion to class separation or consistency, encoding supervised information. The presented ClassNeRV algorithm is based on adjusting a stress minimization objective to balance these two types of distortions while allowing the user to control the amount of supervision considered in the embedding.

Strengths: The presented formalism of missed and false neighbors is insightful, and provides a natural way of controlling the effect of supervised information in the embedding process. This control can be quite useful in data exploration, where different perspectives and patterns in the data can emerge when introducing various supervised meta information. Furthermore, this work also leverages the presented formalism to propose new evaluation criteria for supervised dimensionality reduction by considering intra-class continuity and inter-class trustworthiness, essentially emphasizing an embedding that respects class segmentation and simplifies (to some extent) the separation boundaries between classes. The authors also position the method sufficiently with respect to other dimensionality reduction methods and demonstrate some advantages of their approach in preserving global (class-aware) structure. One of the main advantages of the proposed approach is that relations between classes are still preserved here as much as possible, while still enabling clear decision boundaries to split between classes. This is in contrast to previous methods emphasizing wide separation, which may further facilitate classification (i.e., via much simpler decision boundaries), but obfuscates global structure such as the overlap between classes in the Isolet data or the "equator" region between the hemispheres in the globe example. This distinction can be considered in terms of class separation versus segmentation, and I would recommend further emphasizing it to clarify this important novelty of the proposed approach. It should be noted that some prominent dimensionality reduction methods (for example, LLE and diffusion maps come to mind) are not mentioned or presented here, but on the other hand it is not clear how much they are related or would contribute to the discussion as they are typically fully unsupervised with no incorporation of supervised information.

Weaknesses: One aspect that could clearly be improved in the manuscript is the addition of quantitative comparisons with other methods. The authors do use their supervised versions of continuity and trustworthiness in Figure 4 to evaluate their method compared to others on the two examined datasets. However, the Globe dataset is clearly essentially a toy example, and it is unclear whether the Isolet data is sufficient for proper comparison. It would be good to consider adding some more datasets and quality metrics from supervised dimensionality reduction literature to provide a more thorough evaluation comparable to the ones shown in other work. Another aspect that could be improved is to clarify the insights provided by global structure on the Isolet data. While intuitively the importance of exhibiting relations between classes in the visualization in Figure 5 sounds reasonable and important, it is currently somewhat hypothetical. It would be good to add some downstream analysis clearly demonstrating the insights enabled by ClassNeRV in this case. Finally, on the methodical side, as far as I know, NeRV is rarely considered a standard NLDR method in practice (compared to visualization methods like tSNE, UMAP, and manifold learning methods like Isomap, LLE, and diffusion maps). Further, some other stress minimization approaches exist (such as multidimensional scaling), as well as force directed formulations of tSNE, UMAP, and elastic embedding (essentially having attractive forces between neighbors and repulsive ones otherwise). It seems to me the same ideas used here for supervising NeRV could be applied using such other formulations. Some further discussion and justification for basing the method on the NeRV stress optimization instead of other approaches seems warranted.

Correctness: The method is sound and presented claims are validated empirically. Some additional quantitative evaluation (comparing to other methods) would be appreciated. See previous comments for more details.

Clarity: The paper is well written and easy to read.

Relation to Prior Work: Prior and related work are sufficiently covered. See previous comments for more details.

Reproducibility: Yes

Additional Feedback: *** Updates following author response *** The authors have largely addressed my concern in their response. It is good to see they intend to use the additional page in the camera ready version to provide further discussion and augment their empirical results. While not directly addressing my comments, it is also appreciated that they intend to add more extensive ablation study in the supplement. I should note that I am not sure I clearly see why splitting the tSNE loss would be significantly more difficult than the NeRV loss. However, this is a minor point and it seems the authors will add discussion clarifying it in the camera ready version. In any case, whether the method can or cannot be easily extended to tSNE (and other methods) should not impact the acceptance decision, since being able to generalize it would only strengthen this work rather than detract from it. In conclusion, my recommendation remains to accept this paper, and since this was also my original recommendation, no change is necessary to my original score. === Original review: === See details in strengths and weaknesses portions of the review.


Review 2

Summary and Contributions: Previous dimensionality Reduction (DR) methods aim at separating data of different classes in low dimensional embedding space (supervised) or preserving the data neighborhood structure of original high dimensional embedding (unsupervised). In this paper, ClassNeRV is proposed to ensure better preservation of the data neighborhood structure by utilizing class information. Besides, they used two new class-aware metrics to evaluate their proposed method. Experimental results demonstrated its effectiveness compared to previous supervised and unsupervised methods.

Strengths: 1. The motivation is straightforward. In detail, the proposed ClassNeRV is derived from unsupervised NeRV and further introduce class information to reduce false neighbors between classes and missed neighbors within classes. 2. This paper is well organized and easy to read.

Weaknesses: I have the following critical concerns about this paper: 1. The used datasets are too simple. More complex datasets such as SculptFaces in NeRV paper are required to evaluate the performance of the proposed algorithm. 2. The contribution of this paper is incremental and is not sufficient to be published at NeurIPS. As detailed above, ClassNeRV could be seen as a variation of NeRV through penalizing within-class missed neighbors and between-class false neighbors with class information. Therefore, in my opinion, it is not a significant contribution. 3. According to Sec 3.2, they derived the ClassNeRV Stress Function from NeRV Stress Function by splitting Eq. 2 into within-class and between-class relations. Therefore, there are two additional distinct class-based trade-off parameters $\tau^{\in}$ and $\tau^{\notin}$, and they claimed that when $\tau^{\in}=\tau^{\notin}$, ClassNeRV is an unsupervised method and reduces to the original NeRV. However, Eq. 3 has already utilized class information and thus I disagree that it is unsupervised. Besides, please clarify that it does not reduce to the original NeRV since ClassNeRV uses Bregman divergence. 4. Please elaborate more on the importance of this issue, \ie, preserving the class structures in DR. In detail, the supervised DR methods could be applied to classification by separating classes while the unsupervised methods could be used to visualize high-dimensional data. I'm confused about what cases ClassNeRV makes sense since it does not have not a clear impact as other methods. 5. The authors claimed that they mainly focused on avoiding the distortions of the neighborhood structure that is harmful to the class structure by NeRV. However, according to appendix Table 2, the proposed methods do not have a great improvement to performance. Besides, considered that the proposed methods have two hyper-parameters to control the balance for within classes and between classes, what's the unique advantage of ClassNeRV? Once more hyper-parameters are introduced, the proposed method is limited. In other words, what case do we need to preserve the class structure rather than directly applying the sota unsupervised methods since in many cases we do not have the labeled dataset? 6. An ablation study about different $\tau^{\in}$ and $\tau^{\notin}$ could be considered. It will make the claim of balancing within classes and between classes more convincing.

Correctness: Yes

Clarity: Yes

Relation to Prior Work: Yes

Reproducibility: Yes

Additional Feedback: I have read the authors' response, and decide to slightly upwardly change my overall score.


Review 3

Summary and Contributions: This paper presents a new dimensionality reduction method (ClassNeRV) for data analysis. It aims to preserve both the neighborhood structure and the class separation in the embedding space. The key point of the technique is on the proposed stress function, which is built based on the existing method NeRV, but it additionally considers the distortion within each cluster. Also, two new measurements are proposed to evaluate the distortion of the embedding space on each cluster. Experiments on one synthetic dataset and one real dataset shows the proposed method outperforms than SOTA, in terms of the several measurements considered.

Strengths: This paper is well-written and easy to follow. The idea of considering the distortion on both neighborhood structure and on individual clusters is certainly good. The Figure 1 in the paper presents a nice example on illustrating the reason and motivation behind this work. The method of extending the stress function of NeRV to consider these two kinds of distortion and further using the Bregman divergence appears seems to be novel. The experiment on each dataset is comprehensive, the result of the proposed method outperforms existing methods in terms of several measurements.

Weaknesses: Although the proposed method outperforms the SOTA. My first concern is whether the novelty of this work is enough to meet the standard of NeurIPS. Specifically, the proposed method (ClassNeRV) looks similar to an existing work called ClassiMap in outline (which is also pointed out by the author on Line 37 and 38). ClassiMap also considers the distortion on both neighborhood structure and the clusters and has the same form of stress function. The difference is on the stress function, where ClassNeRV used the one similar to NeRV. Due to this similarity, I think it might be helpful to further illustrate the advantage of the proposed stress function, from either empirical result or a theoretical perspective. This problem can also be greatly alleviated by showing some examples where ClassNeRV can outperform ClassiMap quite a lot either visually or in terms of some measurements. My second concern is about the experiment. This work is proposed to handle the distortion on both neighborhood structure and the cluster structure. While in the dataset used in this work, it is common that these two structures are essentially the same (Like in the Globe dataset the two clusters are just two hemisphere), then I think this example might be a too easy to showcase the ability of the proposed method. So, I was wondering is that possible to show an example, where the neighborhood and the different cluster structures are mixed, like the lower-right part the Figure 1(d)? I would like to see how is the result when the two objectives (preserving neighborhood structure and cluster structure) have some conflict. Some minor questions: (1) Could you provide the expression of the $J_max$ and $e_max$ in Eq.(4) and Eq.(5)? This might be useful to reproduce the result. (2) I think there misses a pair of brackets in Eq.(5), should it be (\rho_{ij} – \kappa) and (r_{ij} - \kappa)? (3) How does the leave-one-out kNN classifier works? Could you please add more description on this? Please let me know if I missed anything, thanks.

Correctness: I think the proposed approach and the empirical methodology are essentially correct.

Clarity: In general, this paper is well-written and easy to follow. The examples and figures are impressive and useful for understanding this work. Several questions about writing have been already mentioned above.

Relation to Prior Work: The related work is discussed comprehensively in this paper. While it might be helpful to add more comparison between the proposed method and ClassiMap, to highlight the novelty and priority of the proposed method.

Reproducibility: No

Additional Feedback: The feedback is appreciated, I would like to keep my score. If possible I think it is helpful to clarify the difference between the proposed method and ClassiMAP in the main paper, and show more comparison betwene these two.


Review 4

Summary and Contributions: A supervised dimensionality reduction and assessment method is proposed. (Checked rebuttal - ok - kept scoring)

Strengths: - well written, scientific sound - established on a NeRV as well supported theoretical method - extensive evaluation - somewhat novel - clear relevant to NeurIPS

Weaknesses: - there is some related work not yet addressed

Correctness: Probably correct

Clarity: Well written

Relation to Prior Work: Yes, but some (very recent and initial) related work is missing

Reproducibility: Yes

Additional Feedback: - overall nice paper not so much to mention - one of the early supervised discrimination methods is missing on page 2 (probably also overfocusing on class separation): Bunte et al. Limited Rank Matrix Learning, discriminative dimension reduction and visualization, Neural Networks, 2012 - 'well-grounded probabilistic framework as NeRV' - well I can understand that a probabilistic model is a nice thing, but please elaborate a bit more why this is particular favourable over other methods from a theoretical / practical point of view --> in particular considering NeRV this is a bit linked to SNE which is somewhat probabilistic as well ... and there is a supervised t-SNE version: Cyril de Bodt, Dounia Mulders, Daniel López Sánchez, Michel Verleysen, John A. Lee: Class-aware t-SNE: cat-SNE. ESANN 2019 - 'We also derive two new class-aware quality indicators' - well there are others measures: J. Venna, J. Peltonen, K. Nybo, H. Aidos, S. Kaski Information retrieval perspective to nonlinear dimensionality reduction for data visualization J. Mach. Learn. Res., 11 (2010), pp. 451-490 L. van der Maaten, E. Postma, H. van den Herik, Dimensionality Reduction: A Comparative Review, Technical Report, Tilburg University Technical Report, TiCC-TR 2009-005, 2009. -- how does it link to your work?

[Author Response · NeurIPS 2020]

We warmly thank the four reviewers for their work and constructive feed-backs. Due to the one page limit, we briefly address here the most crucial comments in groups (R1, R2, R3 and R4 denote concerns raised by the corresponding reviewers). In the revised paper, we will of course do our best to address all reviewers' comments using the additional page allowed for the camera-ready.

**ClassNeRV use cases (R1, R2)** ClassNeRV is designed for exploratory analysis of labeled data, as unsupervised techniques, but steers unavoidable distortions to minimize their impact on classes. Thus, it shows the global structure of classes (class segmentation) rather than extracting features for classification (class separation). It is useful to detect if classes are well-separated or not given a data feature space, to question the labels (meaning of the classes) or the features (feature engineering). For the Isolet data, global structure of ClassNeRV map could help a domain expert discover that, in this feature space, letters are strongly grouped by vowel sounds (BDEGPTV, FS or MN), with a secondary effect of the consonant. This will be clarified in introduction and Isolet interpretation.

**Advantages of Neighbourhood Embedding (NE) family (R3, R4)** NE methods benefit from interesting practical properties such as shift-invariance, making them robust to the curse of dimensionality, which justifies better performances of ClassNeRV compared to Classimap on high dimensional datasets, as observed on Isolet data (Figure 4). On the theoretical side, NE has been explained through a probabilistic framework as a tool for performing a neighbourhood retrieval task in the map [Venna *et al.* 2010], leading to more interpretability for the visual exploration process. **Choice of NeRV among that family (R1)** NeRV is better-suited than other more popular methods such as tSNE due to its divergence penalizing both false and missed neighbours through two independent terms that may be balanced, providing a built-in tunability. We may note that JSE also satisfies those properties, and that we plan to extend the approach to ClassJSE in a longer paper. The reasons of that choice will be incorporated in the section concerning the NE family.

**Hyper-parameters (R2)** Figure 2 illustrates the flexibility of the method by showing its sensitivity to several values of $\tau^\in$ and $\tau^\notin$. Based on that, we then restrict the number of parameters by fixing $\tau^* = 0.5$ and $\epsilon = 0.5$ for the supervised ClassNeRV. An ablation study will be added in supplemental to show individual impact of each of the four components of ClassNeRV stress. **Equivalence of unsupervised ClassNeRV ($\tau^\in = \tau^\notin$) and NeRV (R2)** We detail here (and will add in supplemental) why ClassNeRV with parameters $\tau^\in = \tau^\notin = \tau$, where $\tau \in [0,1]$, is unsupervised and corresponds to NeRV with trade-off parameter $\tau$. In that case, ClassNeRV stress (Equation 3) may be factored by $\tau$ and $(1-\tau)$, so that the sums of within class terms (*i.e.* $\sum_{i,j \in \mathcal{S}_i^\in} ...$) and between class terms (*i.e.* $\sum_{i,j \in \mathcal{S}_i^\notin} ...$) collapse in a sum of all terms that does not take into account the class-information (*i.e.* $\sum_{i,j \in \mathcal{S}_i^\in \cup \mathcal{S}_i^\notin} ... = \sum_{i,j \neq i} ...$), leading to:

$$\zeta_{\text{ClassNeRV}} = \tau \sum_{i,j \neq i} \left( \beta_{ij} \log\left(\frac{\beta_{ij}}{b_{ij}}\right) + b_{ij} - \beta_{ij} \right) + (1-\tau) \sum_{i,j \neq i} \left( b_{ij} \log\left(\frac{b_{ij}}{\beta_{ij}}\right) + \beta_{ij} - b_{ij} \right).$$ Knowing that $\sum_{j \neq i} \beta_{ij} = \sum_{j \neq i} b_{ij} = 1$ (due to the normalization in Equation 1), $\sum_{j \neq i} \beta_{ij}$ and $\sum_{j \neq i} b_{ij}$ cancel each other out, so that $b_{ij} - \beta_{ij}$ and $\beta_{ij} - b_{ij}$ terms may be removed from the above equation. As a result, the Bregman divergence becomes a Kullback-Leibler divergence and ClassNeRV stress equals the stress of NeRV (Equation 3).

**Supplementary datasets and quality indicators (R1, R2, R3, R4)** In the submitted paper, we chose to focus on a few datasets with detailed interpretation and evaluation. We especially resorted to several toy datasets to illustrate limitations of existing unsupervised and supervised DR techniques, as well as the premises of our methodology, which seems to be successful, since all the reviewers well understood our approach. Yet, we fully agree that the confirmatory results obtained with Isolet data are not sufficient, and we would like to benefit from the supplementary page of the camera-ready paper to add results on other high dimensional data. The choice of a well-known image dataset such as the SculptFaces appears very relevant, since it allows representations that intuitively show the true similarities between data points. As this specific dataset does not contain class-information, we propose to use the similar handwritten digits dataset, both with its true labels and with randomly selected labels. The latter provides a case of high dimensional data with conflicting neighbourhood and class structures (as observed in Figure 1d). Example 1 shows a preliminary ClassNeRV map for this dataset, with digits shapes coloured based on the random classes provided to the algorithm. We see that the preservation of neighbourhoods prevails over the preservation of classes, with

Ex. 1: Random labels digits

the fake classes remaining mixed. Some of the many supervised indicators of the literature will be added. Yet, most of them being based on $k$-NN performances in the embedding space [Maaten, Postma, Herrick 2009, Venna *et al.* 2010, de Bodt *et al.* 2019] , they should show the same trends as the $k$-NN accuracy presented in the paper.

**Other issues (R1, R3, R4)** The suggested references will be incorporated. The normalizing terms $\mathcal{T}_{\max}(\kappa) = \mathcal{C}_{\max}(\kappa)$ are given by $\kappa N(2N - 3\kappa - 1)/2$ if $\kappa \leq N/2$ and $N(N - \kappa - 1)/2$ if $\kappa > N/2$ [Venna PhD Thesis]. The leave one-out-classifier (in $k$-NN accuracy) attributes to each point $i$ the majority label (winner takes all strategy) of it $k$ nearest neighbours (among all points except $i$) and the equality case is decided randomly.

[Meta-Review · NeurIPS 2020]

Three referees indicate accept, one indicates that the paper is marginally below threshold. I agree with reviewers 1, 2 and 4 that the presented approach is insightful and useful to NeurIPS applications, and support an accept after reading the rebuttal. However, when revising the paper, please take into account reviewers' concerns about improving quantitative comparisons with other similar methods as well as providing further discussion. Please consider adding experimental support with more complex data to the the main paper or Supplementary Materials.